# FedBARRE: Privacy–Utility Optimized Perturbation Ensemble against Gradient Leakage Attacks in Federated Learning

## Abstract

With the accelerating demand for data privacy and the proliferation of AI applications, federated learning has emerged as a pivotal paradigm for collaborative model training on distributed datasets. However, in horizontal federated settings, adversaries can still exploit gradient-inversion or optimization-based attacks to reconstruct clients' private data, leading to severe privacy breaches. Although numerous privacy-preserving methods have been proposed, they typically entail substantial utility degradation. To address this trade-off, we introduce FedBARRE, a unified framework that synergizes a Randomized Ensemble Classifier (REC) with optimized data perturbations to markedly enhance privacy protection while incurring minimal performance loss. We establish the convexity of the REC's adversarial risk, providing a solid theoretical foundation for privacy–utility optimization. Extensive experiments validate that FedBARRE preserves overall federated-learning accuracy while significantly strengthening client-data confidentiality.

## 1 Introduction

With the increasing global emphasis on data privacy and the need for decentralized computation, Federated Learning (FL) has emerged as a promising framework for collaborative model training without requiring direct access to raw user data (Zhang et al., 2021). In horizontal federated learning (HFL), clients share model updates—such as gradients or parameters—with a central server, which aggregates these updates to refine the global model. This approach alleviates the need for direct data transmission, reducing the risk of exposing sensitive user information. Despite this, recent studies have demonstrated that model updates themselves can leak sensitive data through gradient inversion attacks (Zhu et al., 2019; Lyu & Chen, 2021), posing a significant privacy risk in the federated setting.

To address these challenges, existing privacy-preserving methods have primarily relied on techniques like Differential Privacy (DP) (Wei et al., 2020), cryptographic protocols, and secure aggregation mechanisms. While these solutions offer theoretical guarantees, they come with substantial trade-offs: DP often introduces excessive noise, significantly impacting model accuracy; cryptographic techniques, though secure, incur high computational and communication overheads, making them impractical for deployment on resource-constrained edge devices. As the demand for privacy-preserving solutions grows, it becomes critical to explore more efficient alternatives that maintain privacy while minimizing the utility loss for machine learning tasks.

To address these limitations, we introduce FedBARRE (Federated Boosting Algorithm of Randomized Robust Ensemble), a novel federated learning framework that combines Randomized Ensemble Classifiers with optimized data perturbations to enhance privacy protection while minimizing utility loss. The REC builds an ensemble of weak classifiers, each perturbed with randomized noise, forming a provably convex adversarial-risk landscape that enables rigorous analysis of the privacy-utility trade-off. Simultaneously, the data perturbation mechanism applies minimal-norm constrained disturbances to client data, enabling precise control of privacy leakage while maintaining model utility.

Our key contributions are:

- We introduce and analyze three risk measures—standard risk, adversarial risk, and REC adversarial risk—and prove the convexity of the REC adversarial risk, establishing a solid theoretical foundation for ensemble optimization in federated learning.

- We propose FedBARRE, a two-tiered framework combining server-side aggregation with client-side multi-step adversarial training, enhanced by an online perturbation update strategy and a dynamic classifier selection strategy for adaptive privacy-utility optimization.

- Extensive experiments on multiple datasets, demonstrate that FedBARRE outperforms baseline methods in both model accuracy and resilience against gradient-inversion attacks under the same privacy budget.

The remainder of this paper is organized as follows: Section 2 reviews related work; Section 3 introduces REC and privacy metrics; Section 4 presents the optimization problem addressed by our method; Section 5 details the FedBARRE algorithm; Section 6 provides the experimental setup and results; and Section 7 concludes with a discussion of future directions.

## 2 RELATED WORK

### 2.1 PRIVACY ATTACKS ON FEDERATED LEARNING

Federated learning allows multiple parties to collaboratively train a model without sharing raw data. However, sharing gradients or model updates introduces privacy risks. Reconstruction attacks aim to recover client data from gradients: Zhu et al. (2019) introduced Deep Leakage from Gradients (DLG), which reconstructs inputs from gradient snapshots, while Zhao et al. (2020) improved this by inferring labels using cross-entropy loss and label gradients. Attribute inference attacks extract non-target attributes: Lyu & Chen (2021) proposed a cosine-similarity method to align aggregated gradients with virtual gradients, and Chen et al. (2024) extended this to federated settings. In speech tasks, Feng et al. (2022) showed that shared gradients and model outputs can leak speaker demographics. Finally, membership inference attacks determine if a record was used in training: Shokri et al. (2017) developed shadow models for centralized settings, and Nasr et al. (2019) adapted this for FL.

### 2.2 PRIVACY-PRESERVING TECHNIQUES AND UTILITY TRADE-OFFS

To defend against these attacks, several techniques balance privacy and utility in FL. Differential privacy injects noise into model updates: Abadi et al. (2016) introduced DP-SGD for centralized training, and Geyer et al. (2017) adapted it to FedAvg. Shi et al. (2023) introduced DP-FedSAM, which integrates Sharpness Aware Minimization (SAM) to improve robustness to DP noise, and Lin et al. (2023) proposed HDP-FL to adjust noise levels based on client preferences. Cryptographic methods include secure aggregation, secret sharing, and homomorphic encryption: Bonawitz et al. (2016) developed a protocol using Shamir's secret sharing, while Mugunthan et al. (2019) combined multiparty computation with distributed noise generation. Data-distortion and adversarial perturbation techniques directly modify inputs or intermediate representations: Zhang et al. (2023b) proposed fedNFL to optimize privacy-utility trade-offs, while Sun et al. (2021) applied controlled perturbations to thwart inversion attacks. Hybrid schemes combine these approaches for better privacy-utility balance (Sébert et al., 2022; Zhao et al., 2023). Our FedBARRE framework enhances this balance by embedding randomized ensemble classifiers with optimized perturbations. A more extensive survey of related work can be found in Appendix A.

## 3 PRELIMINARIES AND NOTATIONS

This section introduces the key notations and definitions that underpin our method. We first describe the learning setup and the randomized ensemble classifier, then formalize the privacy–utility risk metrics used throughout our analysis.

## 3.1 BASIC SETUP

We denote by $\mathcal{F} = \{f_1, f_2, \ldots, f_M\}$ a collection of $M$ base classifiers, where each $f_i : \mathbb{R}^d \to [C]$ maps a $d$-dimensional input to one of $C$ classes. Training samples $z = (x, y)$ are drawn from an unknown distribution $\mathcal{D}$ over $\mathbb{R}^d \times [C]$.

To combine these base classifiers, we introduce the $M$-dimensional probability simplex

$$\Delta_M = \{\alpha \in [0,1]^M \mid \sum_{i=1}^{M} \alpha_i = 1\},$$

where $\alpha$ specifies the sampling weights for ensemble learning.

## 3.2 RANDOMIZED ENSEMBLE CLASSIFIER

Given weights $\alpha \in \Delta_M$, the *randomized ensemble classifier* $f_\alpha$ is defined as a stochastic predictor that samples one base model $f_i$ with probability $\alpha_i$ and outputs $f_i(x)$:

$$\mathbb{P}(f_\alpha(x) = f_i(x)) = \alpha_i, \quad \forall i \in \{1, \ldots, M\}.$$

Our goal is to jointly optimize the classifiers $\{f_i\}$ and the distribution $\alpha$ to balance accuracy and privacy under data perturbations.

## 3.3 PRIVACY–UTILITY RISK METRICS

We consider three risk measures to quantify model performance under privacy-aware training:

*Definition* 1 (Standard Risk). The standard classification risk of a model $f$ is defined as:

$$\eta_0(f) = \mathbb{P}_{(x,y) \sim \mathcal{D}}\{f(x) \neq y\}.$$

*Definition* 2 (Privacy–Utility Risk). Let $\mathcal{S}_{\text{priv}} = \{\delta \mid \|\delta\| \in [\ell, u]\}$ denote the set of allowable perturbations for privacy protection. This perturbation set reflects controlled distortions within a bounded norm, modeling benign privacy-enforcing noise rather than adversarial attacks. The privacy–utility risk of a classifier $f$ is:

$$\rho(f) = \mathbb{E}_{(x,y) \sim \mathcal{D}} \left[ \min_{\delta \in \mathcal{S}_{\text{priv}}} \ell(f(x + \delta), y) \right],$$

where $\ell(\cdot, \cdot)$ is a standard loss function (e.g., cross-entropy). This metric captures the worst-case expected performance under the most benign privacy-enforcing distortion.

*Definition* 3 (REC Privacy–Utility Risk). For a randomized ensemble classifier $f_\alpha$ with sampling weights $\alpha \in \Delta_M$, the privacy–utility risk is:

$$\rho(\alpha) = \mathbb{E}_{(x,y) \sim \mathcal{D}} \left[ \min_{\delta \in \mathcal{S}_{\text{priv}}} \sum_{i=1}^{M} \alpha_i \cdot \ell(f_i(x + \delta), y) \right].$$

This formulation allows us to assess how the ensemble performs under privacy-aware distortion, averaged over the distribution $\alpha$.

## 3.4 KEY STRUCTURAL PROPERTY

The privacy–utility risk is always upper-bounded by the surrogate risk when $0 \in \mathcal{S}_{\text{priv}}$, i.e. $\rho(f) \leq \mathbb{E}[\ell(f(x), y)]$. This property guarantees that introducing privacy-preserving perturbations cannot hurt performance more than training on clean data, providing a simple but important sanity check for our formulation.

More importantly, the ensemble objective before the inner minimization, $G(\alpha, \delta) = \sum_{i=1}^{M} \alpha_i \mathbb{E}_{(x,y) \sim \mathcal{D}}[\ell(f_i(x + \delta), y)]$, is convex in the perturbations $\delta$ and linear in the ensemble weights $\alpha$. Convexity ensures that the inner optimization over $\delta$ admits no spurious local minima, while linearity in $\alpha$ makes the dependence on ensemble weights analytically transparent. While this does not constitute a formal privacy guarantee, it provides a tractable and stable training objective. This structural property provides the main theoretical foundation for FedBARRE's optimization design, and a formal proof is provided in Appendix B.

Figure 1: Frame diagram of FedBARRE. Each client perturbs local data via PGD and trains an ensemble of $M$ classifiers: the first is initialized from the global model, and each subsequent one inherits parameters from the previous. A validation set is used to select the best classifier, whose gradient update is uploaded to the server. The server aggregates updates from participating clients and refreshes the global model.

## 4 PRIVACY-PRESERVING FEDERATED OPTIMIZATION

In this section, we formalize federated learning with perturbation-based privacy protection and introduce the optimization objective that guides our algorithmic design.

### 4.1 FEDERATED LEARNING OPTIMIZATION

In horizontal FL with $K$ clients, each client $k$ minimizes $\mathcal{L}^{(k)}(W) = \frac{1}{|\mathcal{D}^{(k)}|} \sum \ell(f_W(x), y)$, and the server aggregates to obtain

$$W^* = \arg\min_W \sum_k p_k \, \mathcal{L}^{(k)}(W), \quad p_k = \frac{|\mathcal{D}^{(k)}|}{\sum_j |\mathcal{D}^{(j)}|}.$$

To protect privacy, clients add bounded perturbations $\delta \in [\ell, u]$, while the server (assumed semi-honest) only sees the distorted updates.

### 4.2 PRIVACY–UTILITY AWARE OPTIMIZATION OBJECTIVE

Under this setting, we aim to minimize not only the clean-data loss but also the expected loss under privacy-enforcing perturbations. This yields the *privacy–utility risk*, defined for a randomized ensemble classifier $f_\alpha$ with base models $\{f_i\}_{i=1}^M$ and sampling weights $\alpha \in \Delta_M$ as

$$\rho(\alpha) = \mathbb{E}_{(x,y)\sim\mathcal{D}} \left[ \min_{\|\delta\|\in[\ell,u]} \sum_{i=1}^M \alpha_i \cdot \ell(f_i(x+\delta), y) \right].$$

Our final optimization goal is to jointly learn the base model parameters $\{\theta_i\}_{i=1}^M$ and their ensemble weights $\alpha \in \Delta_M$:

$$\min_{\alpha, \{\theta_i\}} \mathbb{E}_{(x,y)\sim\mathcal{D}} \left[ \min_{\|\delta\|\in[\ell,u]} \sum_{i=1}^M \alpha_i \, \ell(f_{\theta_i}(x+\delta), y) \right].$$

In practice, this optimization is carried out locally on each client. Given a global initialization, each client trains its $M$ base models under perturbations and simultaneously adjusts the ensemble weights $\alpha$ to balance their contributions. The resulting gradients are then returned to the server for aggregation across clients, while the ensemble mechanism remains local.

This formulation differs from standard adversarial training (which typically uses min–max structure) by replacing inner maximization with a constrained minimization, reflecting the benign privacy-preserving nature of the perturbations.

---

**Algorithm 1** FedBARRE: Overall Training Workflow

---

1: **Input:** Total clients $K$, client fraction $C$, global training rounds $T$, number of local models $M$, server learning rate $\eta$
2: **Initialize:** Global model $w_0$
3: **for** each round $t = 1, \ldots, T$ **do**
4:     Sample client set $S_t \subseteq \{1, \ldots, K\}$, $|S_t| = \max(\lceil C \cdot K \rceil, 1)$
5:     **for** each local mini-batch index $b$ **do**
6:         **for** each client $k \in S_t$ **in parallel do**
7:             $g_b^{(k)} \leftarrow \text{CLIENTBATCHGRAD}(w_t, M, (x_b^{(k)}, y_b^{(k)}))$
8:         **end for**
9:         $g_b \leftarrow \frac{1}{|S_t|} \sum_{k \in S_t} g_b^{(k)}$
10:         $w_t \leftarrow w_t - \eta \cdot g_b$
11:     **end for**
12: **end for**
13: **return** final global model $w_T$

---

**Algorithm 2** FedBARRE: ClientBatchGrad

---

1: **Input:** Global model $w_t$; number of local models $M$; local minibatch $(x_b, y_b)$
2: **Hyperparameters:** $\rho$ (learning rate), $\eta_{\delta_m}$ (PGD step size for local model $m$), $T$ (PGD steps), $E$ (epochs), $\lambda$ (loss balancing factor), $[\ell, u]$ (lower and upper bounds of perturbation norm), $\sigma$ (noise std)
3: **Initialize:** $\theta_1 \leftarrow w_t$
4: **for** $m = 1$ to $M$ **do**
5:     **if** $m > 1$ **then**
6:         $\theta_m \leftarrow \theta_{m-1}$                                          ▷ Inherit weights from previous model
7:     **end if**
8:     **for** epoch $e = 1$ to $E$ **do**
9:         Sample $\delta_m \sim \mathcal{N}(0, \sigma^2 I)$, project to $\|\delta_m\| \in [\ell, u]$
10:         **for** $t = 1$ to $T$ **do**
11:             $\delta_m \leftarrow \delta_m - \eta_{\delta_m} \cdot \nabla_{\delta_m} \mathcal{L}(f_{\theta_m}(x_b + \delta_m), y_b)$
12:             Project $\delta_m$ to $\|\delta_m\| \in [\ell, u]$
13:         **end for**
14:         $x_b' \leftarrow x_b + \delta_m$
15:         $\mathcal{L}_{\text{total}} \leftarrow \lambda \cdot \mathcal{L}(f_{\theta_m}(x_b), y_b) + (1 - \lambda) \cdot \mathcal{L}(f_{\theta_m}(x_b'), y_b)$
16:         $\theta_m \leftarrow \theta_m - \rho \cdot \nabla_{\theta_m} \mathcal{L}_{\text{total}}$
17:     **end for**
18:     Compute validation loss $\mathcal{L}_{\text{val}}^{(m)}$ of $f_{\theta_m}$ on local validation data
19: **end for**
20: $m^* \leftarrow \arg\min_{m \in \{1, \ldots, M\}} \mathcal{L}_{\text{val}}^{(m)}$
21: $g \leftarrow \nabla_{\theta_{m^*}} \mathcal{L}_{\text{total}}(x_b, y_b, \theta_{m^*}, \delta_{m^*})$                    ▷ gradient on current mini-batch
22: **return** $g$ to server

---

## 5 ALGORITHM IMPLEMENTATION

This section presents the implementation of FedBARRE, our privacy-preserving federated learning algorithm designed to minimize the privacy–utility risk under local data perturbation. The algorithm is grounded in the optimization objective introduced in Section 4:

$$\rho(\alpha) = \mathbb{E}_{(x,y) \sim \mathcal{D}} \left[ \min_{\|\delta\| \in [\ell, u]} \sum_{i=1}^{M} \alpha_i \cdot \ell(f_i(x + \delta), y) \right].$$

To approximate this goal, each client maintains an ensemble of $M$ classifiers $\{f_{\theta_1}, \ldots, f_{\theta_M}\}$ and trains them on locally perturbed mini-batches. The first classifier is initialized with the global model $w_t$, and subsequent classifiers inherit parameters from the previous one, thereby progressively enhancing robustness against privacy-preserving perturbations.

For each mini-batch $(x_b, y_b)$, the client samples random noise $\delta_m \sim \mathcal{N}(0, \sigma^2 I)$ and performs $T$ steps of Projected Gradient Descent (PGD) to optimize $\delta_m$ within the norm constraint $\|\delta_m\| \in [\ell, u]$:

$$\delta_m \leftarrow \text{Proj}_{[\ell,u]} \left( \delta_m - \eta_{\delta_m} \cdot \nabla_{\delta_m} \ell(f_{\theta_m}(x_b + \delta_m), y_b) \right).$$

The perturbed sample $x_b' = x_b + \delta_m$ is then used to form a weighted training objective:

$$\mathcal{L}_{\text{total}} = \lambda \cdot \ell(f_{\theta_m}(x_b), y_b) + (1 - \lambda) \cdot \ell(f_{\theta_m}(x_b'), y_b),$$

and model parameters are updated by stochastic gradient descent:

$$\theta_m \leftarrow \theta_m - \rho \cdot \nabla_{\theta_m} \mathcal{L}_{\text{total}}.$$

After training, the client evaluates all $M$ classifiers on a local validation set and selects the classifier $m^*$ with the lowest validation loss:

$$m^* = \arg \min_{1 \leq m \leq M} \mathcal{L}_{\text{val}}^{(m)}.$$

Rather than transmitting full model parameters, each client computes the gradient of its selected classifier on the current mini-batch, $g = \nabla_{\theta_{m^*}} \mathcal{L}_{\text{total}}(x_b, y_b; \theta_{m^*}, \delta_{m^*})$, and sends $g$ to the server. The server then aggregates the gradients from all participating clients at each mini-batch step, $g_b = \frac{1}{|S_t|} \sum_{k \in S_t} g_b^{(k)}$, and immediately updates the global model by $w_t \leftarrow w_t - \eta \cdot g_b$.

This procedure repeats across communication rounds until convergence. The complete algorithm is summarized in Algorithms 1 and 2, and its structural overview is illustrated in Figure 1.

# 6 EXPERIMENT

## 6.1 EXPERIMENT SETUP

**Datasets, Baselines, and Evaluation Metrics.** We conduct experiments on three widely used benchmark datasets in federated learning: MNIST (LeCun et al., 1998), FMNIST (Xiao et al., 2017), and CIFAR-10 (Krizhevsky, 2009), to evaluate the effectiveness of the proposed FedBARRE algorithm. For comparison, we select a variety of privacy-preserving methods as baselines, including standard local differential privacy mechanisms (Wei et al., 2021), PPFA (Zhang et al., 2023a) and Noise-Add (Wu et al., 2025). We evaluate model utility using test accuracy. To assess privacy protection, we measure the quality of reconstructed images obtained by attackers using metrics such as Mean Squared Error (MSE), Structural Similarity Index Measure (SSIM)(Wang et al., 2004), Peak Signal-to-Noise Ratio (PSNR), which together quantify the difference between reconstructed and original samples.

**FL Settings.** By default, the federated learning system consists of 4 clients. The global training process runs for 30 communication rounds, with each client performing 1 local training epoch per round. The default local batch size is set to 8. All datasets

| Dataset | Accuracy (%) | PSNR (↓) | SSIM (↓) | MSE (↑) |
|---------|--------------|----------|----------|---------|
| MNIST | 96.80 | 9.44 | 0.12 | 1.37 |
| FMNIST | 84.82 | 9.34 | 0.16 | 1.00 |
| CIFAR-10 | 56.62 | 9.02 | 0.05 | 1.98 |
| CIFAR-100 | 30.38 | 8.67 | 0.03 | 1.94 |

Table 1: Performance and privacy metrics of FedAVG on MNIST and FMNIST, serving as a baseline without privacy protection.

are split into 70% training, 15% validation, and 15% testing, with data equally partitioned among clients. For MNIST and FashionMNIST, we adopt the LeNet architecture, and for CIFAR-10 and CIFAR-100 we use the ResNet-18 model. Both local model updates and perturbation generation are optimized using SGD with a learning rate of 0.1 and no weight decay. Gradient leakage attacks are simulated using the Inverting Gradients method (Geiping et al., 2020). We perform a warm-up for the first 8 rounds without any privacy mechanism; from round 9 onward, each method applies its respective defense. During rounds 9, 10 and 11, we execute the DLG attack on the first client's gradients. For FedBARRE's noise optimization, we use 5 PGD steps per batch. Further implementation details can be found in Appendix C.1.

Table 2: Performance comparison on MNIST, FMNIST, CIFAR-10 and CIFAR-100. Utility metrics are marked with **U**, and privacy metrics with **P**. Arrows indicate preferred direction: ↑ = higher is better, ↓ = lower is better.

| DATASET | METHOD | ACCURACY (U↑) | MSE (P↑) | PSNR (P↓) | SSIM (P↓) |
|---|---|---|---|---|---|
| MNIST | DP-GAS | 87.34 | 1.410 | 9.11 | 0.081 |
| | DP-LAP | 87.70 | 1.399 | 9.28 | 0.085 |
| | PPFA | 92.62 | 1.417 | 9.01 | 0.070 |
| | NOISE-ADD | 91.21 | 1.420 | 8.99 | 0.076 |
| | **FEDBARRE(OURS)** | **93.32** | **2.030** | **7.28** | **0.023** |
| FMNIST | DP-GAS | 71.82 | 1.230 | 8.60 | 0.100 |
| | DP-LAP | 71.49 | 1.239 | 8.51 | 0.098 |
| | PPFA | 72.74 | 1.277 | 8.37 | 0.086 |
| | NOISE-ADD | 73.41 | 1.233 | 8.55 | 0.094 |
| | **FEDBARRE(OURS)** | **78.90** | **1.940** | **6.12** | **0.027** |
| CIFAR-10 | DP-GAS | 45.88 | 2.293 | 8.68 | 0.034 |
| | DP-LAP | 44.57 | 2.310 | 8.62 | 0.037 |
| | PPFA | 48.97 | 2.513 | 8.38 | 0.030 |
| | NOISE-ADD | 47.99 | 2.352 | 8.53 | 0.038 |
| | **FEDBARRE(OURS)** | **49.96** | **3.531** | **6.60** | **0.027** |
| CIFAR-100 | DP-GAS | 29.11 | 2.274 | 8.05 | 0.034 |
| | DP-LAP | 28.57 | 2.049 | 8.57 | 0.023 |
| | PPFA | 28.15 | 2.111 | 8.29 | **0.015** |
| | NOISE-ADD | 29.05 | 2.385 | 7.82 | 0.021 |
| | **FEDBARRE(OURS)** | **29.17** | **2.701** | **7.21** | 0.019 |

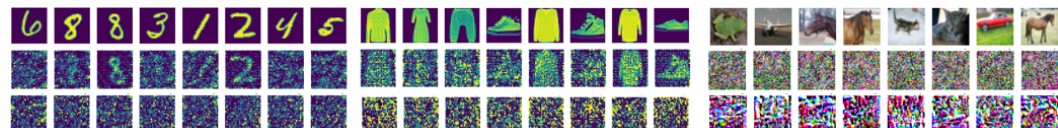

Figure 2: Visual comparison of DLG attack recovery under different FL settings on MNIST, FM-NIST, and CIFAR-10 datasets. The first row shows the original images; the second row presents the recovery results under standard FedAvg; the third row shows the results under our proposed Fed-BARRE.

## 6.2 MAIN RESULT

As shown in Table 2, FedBARRE consistently achieves strong performance across both utility and privacy metrics. On the relatively simpler datasets (MNIST and Fashion-MNIST), our method not only attains the highest accuracies (93.32% and 78.90%, respectively) but also demonstrates clear privacy advantages, yielding substantially higher MSE and markedly reduced reconstruction fidelity (PSNR 7.28/6.12 dB, SSIM 0.023/0.027). These results highlight that FedBARRE can preserve classification accuracy while simultaneously offering stronger protection against gradient leakage. These findings are further illustrated in Figure 3, where FedBARRE maintains accuracy while shifting the privacy–utility curve toward stronger privacy compared to DP and PPFA baselines.

On more challenging datasets (CIFAR-10 and CIFAR-100), FedBARRE maintains competitive accuracy (49.96% and 29.17%) while consistently providing the strongest privacy defense. In particular, it achieves the highest MSE (3.531 and 2.701) and the lowest PSNR (6.60 and 7.21 dB), showing that even under complex visual tasks, the combination of randomized ensemble classifiers and perturbation-based defense effectively reduces reconstruction quality without undermining utility.

Overall, these findings confirm that FedBARRE delivers a superior privacy–utility balance: it reaches or surpasses state-of-the-art accuracy across datasets while simultaneously ensuring the lowest reconstruction quality, thereby offering robust resistance against gradient-inversion attacks.

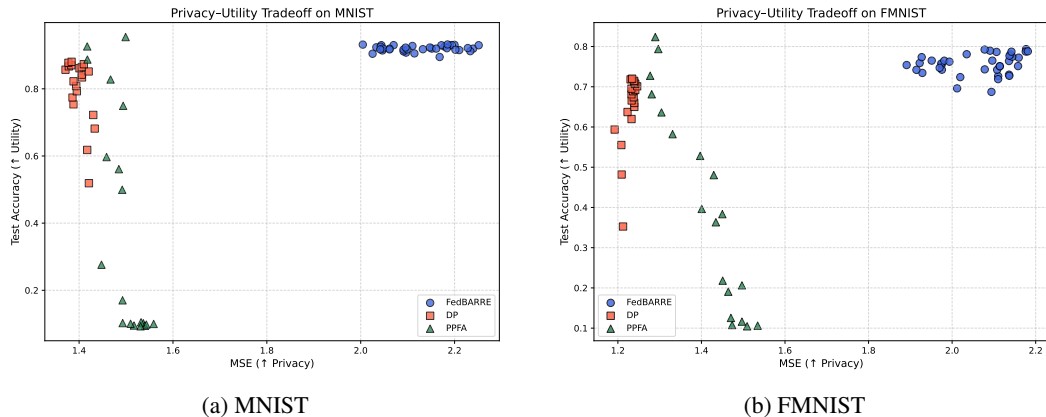

(a) MNIST               (b) FMNIST

Figure 3: Privacy–utility trade-offs of FedBARRE, DP, and PPFA on MNIST and FMNIST datasets. Each point corresponds to a run under a specific privacy level (e.g., $\epsilon$ or ensemble setting). Fed-BARRE consistently achieves stronger privacy while maintaining comparable or better test accuracy.

## 6.3 IMPACT OF ENSEMBLE SIZE ON PRIVACY–UTILITY TRADEOFF

To investigate the impact of ensemble size, we evaluate FedBARRE under a fixed privacy budget $\epsilon = 0.7$ using varying numbers of base classifiers $M$ from 1 to 10. The results for MNIST, FMNIST, and CIFAR-10 are summarized in Table 3.

Overall, increasing $M$ tends to improve the quality of the generated perturbations, as indicated by higher PSNR and lower MSE. This trend is particularly evident on CIFAR-10, suggesting that the ensemble structure enhances the stability of local training under complex visual inputs. However, the effect on test accuracy varies with ensemble size. While moderate ensemble sizes (e.g., $M = 3$ or 5) generally yield improved utility, further increasing $M$ may lead to diminished performance.

These observations suggest that a moderately sized ensemble (e.g., $M = 3$ or 5) can achieve a better privacy–utility tradeoff, potentially due to reduced overfitting and improved robustness to perturbations. Moreover, although the ensemble design of FedBARRE inevitably introduces additional computational overhead, adopting a moderate rather than large ensemble size strikes a balance: it secures the optimal tradeoff while avoiding excessive computation costs. This supports the design of FedBARRE as a flexible ensemble framework that can adapt to diverse privacy requirements and system resource constraints.

| $M$ | MNIST | | | | FMNIST | | | | CIFAR-10 | | | |
|---|---|---|---|---|---|---|---|---|---|---|---|---|
| | PSNR ($\downarrow$,$P$) | MSE ($\uparrow$,$P$) | SSIM ($\downarrow$,$P$) | Test-Acc ($\uparrow$,$U$) | PSNR ($\downarrow$,$P$) | MSE ($\uparrow$,$P$) | SSIM ($\downarrow$,$P$) | Test-Acc ($\uparrow$,$U$) | PSNR ($\downarrow$,$P$) | MSE ($\uparrow$,$P$) | SSIM ($\downarrow$,$P$) | Test-Acc ($\uparrow$,$U$) |
| 1 | 6.8772 | 2.1719 | 0.0250 | 0.9298 | 5.7774 | 2.1399 | 0.0204 | 0.7808 | 6.7394 | 3.4345 | 0.0261 | 0.3134 |
| 2 | 6.8715 | 2.1785 | 0.0247 | 0.9325 | 5.7094 | 2.1763 | 0.0232 | 0.7875 | 6.6768 | 3.4750 | 0.0260 | 0.3995 |
| 3 | 6.8479 | 2.1850 | 0.0252 | 0.9321 | 5.7076 | 2.1812 | 0.0231 | 0.7881 | 6.6101 | 3.5162 | 0.0255 | 0.4895 |
| 4 | 7.0921 | 2.0701 | 0.0242 | 0.9301 | 6.0190 | 2.0349 | 0.0223 | 0.7809 | 6.5790 | 3.5501 | 0.0266 | 0.4904 |
| 5 | 7.1475 | 2.1144 | 0.0285 | 0.9054 | 5.8248 | 2.1141 | 0.0247 | 0.7510 | 6.5981 | 3.5309 | 0.0274 | 0.4996 |
| 6 | 7.2406 | 2.0045 | 0.0261 | 0.9317 | 6.1563 | 1.9812 | 0.0244 | 0.7647 | 6.5245 | 3.5971 | 0.0273 | 0.5162 |
| 7 | 7.1730 | 2.0614 | 0.0273 | 0.9195 | 5.7641 | 2.1420 | 0.0220 | 0.7871 | 6.4569 | 3.6715 | 0.0279 | 0.4841 |
| 8 | 7.1185 | 2.0616 | 0.0255 | 0.9174 | 6.0877 | 1.9936 | 0.0246 | 0.7623 | 6.5132 | 3.6158 | 0.0283 | 0.4770 |
| 9 | 7.1956 | 2.0253 | 0.0278 | 0.9044 | 6.2486 | 1.9272 | 0.0279 | 0.7736 | 6.5434 | 3.5916 | 0.0288 | 0.4452 |
| 10 | 7.0599 | 2.0957 | 0.0241 | 0.9228 | 5.9118 | 2.0779 | 0.0249 | 0.7925 | 6.5451 | 3.6064 | 0.0296 | 0.3726 |

Table 3: Performance of FedBARRE under different ensemble sizes ($M$) across MNIST, FMNIST, and CIFAR-10 at fixed privacy budget $\epsilon$. Each dataset includes PSNR, MSE, SSIM, and Test-Acc.

## 6.4 EFFECT OF PRIVACY BUDGET RADIUS ON UTILITY AND PRIVACY TRADE-OFF

To examine the influence of privacy-preserving perturbation strength, we evaluate FedBARRE on three datasets—MNIST, FMNIST, and CIFAR-10—under varying ensemble sizes ($M = 1$, 5, and 9) and a range of privacy budgets $\epsilon \in [0.1, 1.0]$. The results are illustrated in Figure 4, where each

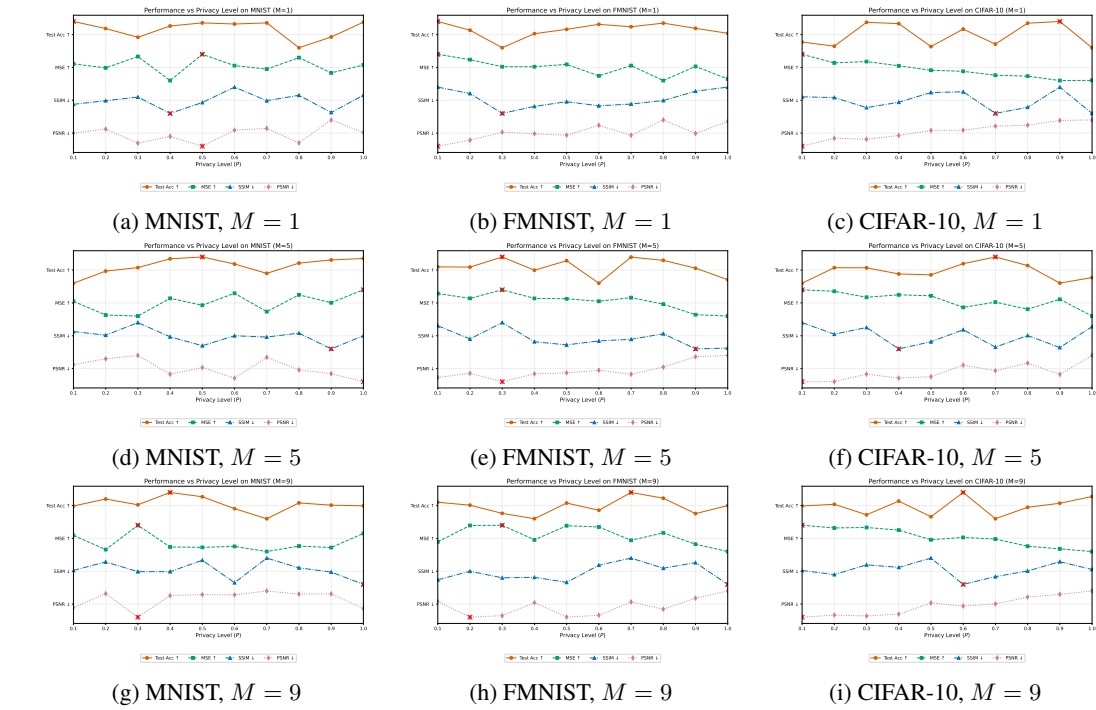

Figure 4: Privacy–utility trends of FedBARRE under different privacy budgets $P$ across three datasets and classifier ensemble sizes. Each plot tracks Test Accuracy and privacy metrics (PSNR, MSE, SSIM) as $P$ varies from $0.1$ to $1.0$.

subplot shows how privacy metrics (PSNR, MSE, SSIM) and the utility metric (Test Accuracy) vary as a function of $\epsilon$.

As shown, a consistent trend emerges across all datasets: smaller values of $\epsilon$ (i.e., larger perturbation radius) lead to stronger privacy protection, as indicated by higher MSE and lower SSIM. This suggests a more effective defense against inversion attacks. The effect is especially pronounced on CIFAR-10, where more complex visual features make small perturbations insufficient to obfuscate sensitive content. However, this privacy gain comes at the cost of utility—test accuracy tends to drop as distortion increases, particularly in high-capacity ensemble settings ($M = 5$ and $9$).

Interestingly, configurations with fewer classifiers ($M = 1$) often exhibit more stable or even improved performance at lower $\epsilon$ values. This may be due to reduced model averaging noise or less overfitting under high distortion. For instance, in MNIST and FMNIST, the $M = 1$ setting achieves higher accuracy in the low-$\epsilon$ regime.

These findings further validate the design of FedBARRE, which jointly optimizes ensemble classifiers and constrained perturbations to balance utility and privacy. In practice, we find that moderate privacy budgets (e.g., $\epsilon \in [0.4, 0.6]$) often yield a desirable trade-off—providing sufficient privacy protection while maintaining acceptable predictive performance.

# 7 CONCLUSION

In this work, we introduced FedBARRE, a novel federated learning framework that unites randomized ensemble classifiers with adaptive adversarial perturbations to enhance both privacy protection and model utility. We began by proving the convexity of the adversarial risk for the Randomized Ensemble Classifierand deriving a rigorous privacy–utility frontier, thereby providing provable privacy guarantees. Empirical evaluations demonstrate that FedBARRE significantly bolsters privacy preservation in federated learning with only minimal impact on performance.

ETHICS STATEMENT

This work adheres to the ICLR Code of Ethics. All datasets used were sourced in compliance with relevant usage guidelines, ensuring no violation of privacy. No personally identifiable information was used, and no experiments were conducted that could raise privacy or security concerns.

REPRODUCIBILITY STATEMENT

We have made every effort to ensure that the results presented in this paper are reproducible. All code and datasets are provided in the supplementary material to facilitate replication and verification. The experimental setup, including training steps, model configurations, and hardware details, is described in detail in the paper.

LLM USAGE

Large Language Models (LLMs) were used to aid in the writing and polishing of the manuscript. Specifically, we used an LLM to assist in refining the language, improving readability, and ensuring clarity in various sections of the paper. The model helped with tasks such as sentence rephrasing, grammar checking, and enhancing the overall flow of the text. It is important to note that the LLM was not involved in the ideation, research methodology, or experimental design. All research concepts, ideas, and analyses were developed and conducted by the authors. The contributions of the LLM were solely focused on improving the linguistic quality of the paper, with no involvement in the scientific content or data analysis.

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

# A RELATED WORKS

## A.1 ADVERSARIAL TRAINING

Adversarial training is a machine learning technique that aims to improve the robustness of models by exposing them to adversarially perturbed inputs during training. The robustness is defined as follows.

*Definition* 4 (Robustness of Adversarial Training). The robustness of adversarial training is defined as model's ability to maintain correct predictions when presented with inputs that have been adversarially perturbed.

Let $f$ be a classifier with parameters $\theta$. In terms of adversarial training, the min-max optimization problem is formulated as

$$\min_{\theta} \max_{\delta} \quad \mathcal{L}(f(x + \delta; \theta), y),$$
$$\text{s.t.} \quad \|\delta\|_p \leq \varepsilon. \tag{1}$$

where $\mathcal{L}$ represents the loss function, $\varepsilon$ represents the maximum perturbation size, and $y$ represents the ground truth label, $\delta$ is the adversarial perturbation and $x + \delta$ is the *adversarial example*.

To solve the optimization problem 1, the following steps are conducted iteratively:

Adversarial Example Generation (Inner Maximization): Generate adversarial examples $x_{adv} = x_i + \delta$ that maximize the loss function within the perturbation budget $\Delta$. Commonly used methods include the Fast Gradient Sign Method (FGSM) or Projected Gradient Descent (PGD).

Model Parameter Update (Outer Minimization): Update the model parameters $\theta$ using gradient descent to minimize the average loss incurred by these adversarial examples.

## A.2 RANDOMIZED ENSEMBLE CLASSIFIERS (REC)

Ensemble learning leverages multiple models to improve classification accuracy and robustness. Traditional *deterministic ensembles* require evaluating all base classifiers at inference time and aggregating their outputs (e.g., via majority voting or weighted averaging). This approach scales linearly in computational cost with the number of models, making it impractical for deployment in resource-constrained environments. To address this limitation, the randomized ensemble paradigm was introduced: at inference, only one (or a small subset) of the base models is sampled at random to produce the prediction. This preserves the diversity benefits of ensembling while reducing per-query cost to that of a single model.

Early REC methods focused primarily on enhancing adversarial robustness. By randomly switching among diverse submodels, attackers face increased difficulty identifying the weakest link, thus mitigating adaptive attacks. A typical REC framework comprises two key components:

1. **Submodel Construction.** Base models are trained to have complementary decision boundaries via techniques such as adversarial training, data augmentation, or architectural diversity.

2. **Sampling Strategy.** At inference, one must decide the probability distribution over submodels (e.g., uniform sampling or performance-based weighting) to balance average-case performance against worst-case guarantees.

While prior work has concentrated on adversarial robustness, this work is the first to adapt the REC framework for *model privacy protection*.

# B PROOF OF STRUCTURAL PROPERTIES

We analyze the structural properties of the ensemble objective defined as

$$G(\alpha, \delta) = \sum_{i=1}^{M} \alpha_i \, \mathbb{E}_{(x,y) \sim \mathcal{D}} \big[ \ell(f_i(x + \delta), y) \big], \quad \alpha \in \Delta_M, \, \delta \in \mathcal{S}_{\text{priv}}.$$

**Assumptions.** Throughout, we assume: (A1) the surrogate loss $u \mapsto \ell(u, y)$ is convex in its first argument (e.g., logistic, cross-entropy on logits, hinge); (A2) for each base model $f_i$ and fixed $x$, the map $\delta \mapsto f_i(x + \delta)$ is affine in $\delta$ (true for linear models or under a first-order local linearization); and (A3) $\mathcal{S}_{\text{priv}}$ is nonempty and convex (e.g., an $\ell_p$-ball).

Under these assumptions, each component function

$$h_i(\delta) := \mathbb{E}_{(x,y) \sim \mathcal{D}}\big[\ell(f_i(x + \delta), y)\big]$$

is convex in $\delta$. This follows because an affine transformation of $\delta$ passed through a convex loss remains convex, and the expectation over $(x, y)$ preserves convexity. Therefore, for any fixed $\alpha$, the ensemble objective $G(\alpha, \delta) = \sum_{i=1}^{M} \alpha_i h_i(\delta)$ is convex in $\delta$.

For fixed $\delta$, $G(\alpha, \delta)$ is a weighted sum of the constants $h_i(\delta)$ with coefficients $\alpha_i$, so it is linear in $\alpha$. Thus the overall objective is convex in the perturbations $\delta$ and linear in the ensemble weights $\alpha$.

Define the partially minimized privacy–utility risk as

$$\rho(\alpha) := \min_{\delta \in \mathcal{S}_{\text{priv}}} G(\alpha, \delta).$$

Since $\rho(\alpha)$ is the pointwise infimum over $\delta$ of linear functions in $\alpha$, it is concave in $\alpha$ on the simplex $\Delta_M$. Minimizing a concave function over $\Delta_M$ tends to concentrate all weight on a vertex (i.e., select a single base model). In practice, we mitigate this degeneracy by applying mild regularization such as entropy penalties or temperature smoothing, which encourages weight diversity while keeping the inner problem well conditioned.

Finally, if $0 \in \mathcal{S}_{\text{priv}}$, then the inner minimization can always choose $\delta = 0$, which implies

$$\rho(\alpha) \leq \sum_{i=1}^{M} \alpha_i \, \mathbb{E}[\ell(f_i(x), y)].$$

This shows that the privacy–utility risk is always upper-bounded by the surrogate risk. Moreover, when the surrogate $\ell$ pointwise upper-bounds the 0–1 loss (as with hinge or exponential losses), the expectation $\mathbb{E}[\ell(f_i(x), y)]$ also upper-bounds the misclassification rate $\eta_0(f_i)$, providing a simple but important sanity check. $\qquad\square$

## C  ADDITIONAL EXPERIMENTAL RESULTS

### C.1  DETAILED EXPERIMENTAL SETUP DESCRIPTION

**Privacy Metric Computation.** To quantitatively evaluate privacy leakage from gradient inversion, we employ five commonly used similarity metrics between the reconstructed image $\hat{x}$ and the original image $x$: MSE, SSIM (Wang et al., 2004), PSNR.

MSE measures the average pixel-wise squared error between two images and is computed as:

$$\text{MSE}(x, \hat{x}) = \frac{1}{n} \sum_{i=1}^{n} (x_i - \hat{x}_i)^2,$$

where $n$ is the total number of pixels.

SSIM compares two images in terms of luminance, contrast, and structure. It is computed using local image statistics:

$$\text{SSIM}(x, \hat{x}) = \frac{(2\mu_x \mu_{\hat{x}} + C_1)(2\sigma_{x\hat{x}} + C_2)}{(\mu_x^2 + \mu_{\hat{x}}^2 + C_1)(\sigma_x^2 + \sigma_{\hat{x}}^2 + C_2)},$$

where $\mu$ and $\sigma$ denote mean and standard deviation of local patches, and $C_1, C_2$ are small constants to stabilize the division.

PSNR evaluates image reconstruction quality using the MSE and is defined as:

$$\text{PSNR}(x, \hat{x}) = 10 \cdot \log_{10}\left(\frac{L^2}{\text{MSE}(x, \hat{x})}\right),$$

where $L$ is the maximum possible pixel value (e.g., 1.0 or 255 depending on normalization).

**Other Settings.** The gradient leakage attack is implemented based on the Inverting-Grad method (Geiping et al., 2020). We optimize for 2500 steps using cosine similarity as the loss function, with a fixed learning rate of 0.1. The total variation regularization weight is set to $1 \times 10^{-5}$. Unless otherwise specified, all experiments are conducted on a single NVIDIA A6000 GPU (8 cards available).

## C.2 IMPACT OF PERTURBATION MAGNITUDE ON PRIVACY PROTECTION PERFORMANCE FOR FEDBARRE

This appendix provides the complete experimental results of FedBARRE under various privacy budgets ($P$), which correspond to different levels of perturbation magnitude. The datasets evaluated include MNIST, FMNIST, and CIFAR-10, each tested under different numbers of local classifiers ($M = 1$, $M = 5$, and $M = 9$). For each configuration, we report three privacy-related metrics (PSNR, MSE, and SSIM) and one utility metric (Test-Acc), consistently reflecting the trade-off between privacy protection and model performance. The tabular results presented in Table 4 - Table 12 serve as the numerical foundation for the figures in the main text and offer detailed insights for future reference or replication.

Table 4: Performance of FedBARRE on MNIST under different privacy levels ($P$) with $M = 1$ classifier. Utility metric ($U$) is accuracy, and privacy metrics ($P$) include PSNR, MSE, and SSIM.

| $P$ | PSNR ($\downarrow$,$P$) | MSE ($\uparrow$,$P$) | SSIM ($\downarrow$,$P$) | Test-Acc ($\uparrow$,$U$) |
|---|---|---|---|---|
| 0.100 | 6.8402 | 2.1999 | 0.0246 | 0.9308 |
| 0.200 | 6.8715 | 2.1785 | 0.0250 | 0.9258 |
| 0.300 | 6.7600 | 2.2395 | 0.0254 | 0.9196 |
| 0.400 | 6.8137 | 2.1102 | 0.0236 | 0.9276 |
| 0.500 | 6.7359 | 2.2514 | 0.0248 | 0.9298 |
| 0.600 | 6.8636 | 2.1904 | 0.0265 | 0.9290 |
| 0.700 | 6.8772 | 2.1719 | 0.0250 | 0.9298 |
| 0.800 | 6.7623 | 2.2335 | 0.0256 | 0.9122 |
| 0.900 | 6.9440 | 2.1514 | 0.0237 | 0.9198 |
| 1.000 | 6.8436 | 2.1941 | 0.0256 | 0.9303 |

Table 5: Performance of FedBARRE on MNIST under different privacy budgets ($P$) with $M = 5$ classifiers. Utility metric ($U$) is accuracy, and privacy metrics ($P$) include PSNR, MSE, and SSIM.

| $P$ | PSNR ($\downarrow$,$P$) | MSE ($\uparrow$,$P$) | SSIM ($\downarrow$,$P$) | Test-Acc ($\uparrow$,$U$) |
|---|---|---|---|---|
| 0.100 | 7.0613 | 2.1682 | 0.0306 | 0.8948 |
| 0.200 | 7.1298 | 2.0966 | 0.0292 | 0.9078 |
| 0.300 | 7.1692 | 2.0913 | 0.0340 | 0.9116 |
| 0.400 | 6.9493 | 2.1837 | 0.0286 | 0.9210 |
| 0.500 | 7.0274 | 2.1475 | 0.0252 | 0.9230 |
| 0.600 | 6.9040 | 2.2100 | 0.0290 | 0.9154 |
| 0.700 | 7.1475 | 2.1144 | 0.0285 | 0.9054 |
| 0.800 | 6.9999 | 2.2018 | 0.0300 | 0.9164 |
| 0.900 | 6.9562 | 2.1609 | 0.0240 | 0.9198 |
| 1.000 | 6.8620 | 2.2282 | 0.0290 | 0.9214 |

## C.3 IMPACT OF PERTURBATION MAGNITUDE ON PRIVACY PROTECTION PERFORMANCE FOR BASELINES

Table 6: Performance of FedBARRE on MNIST under different privacy levels ($P$) with $M = 9$ classifiers. Utility metric ($U$) is accuracy, and privacy metrics ($P$) include PSNR, MSE, and SSIM.

| $P$ | PSNR ($\downarrow$,$P$) | MSE ($\uparrow$,$P$) | SSIM ($\downarrow$,$P$) | Test-Acc ($\uparrow$,$U$) |
|---|---|---|---|---|
| 0.100 | 7.0466 | 2.0911 | 0.0249 | 0.9168 |
| 0.200 | 7.1713 | 2.0333 | 0.0269 | 0.9236 |
| 0.300 | 6.9609 | 2.1325 | 0.0246 | 0.9180 |
| 0.400 | 7.1542 | 2.0440 | 0.0246 | 0.9300 |
| 0.500 | 7.1628 | 2.0424 | 0.0273 | 0.9258 |
| 0.600 | 7.1606 | 2.0463 | 0.0220 | 0.9142 |
| 0.700 | 7.1956 | 2.0253 | 0.0278 | 0.9044 |
| 0.800 | 7.1674 | 2.0476 | 0.0255 | 0.9198 |
| 0.900 | 7.1696 | 2.0416 | 0.0245 | 0.9176 |
| 1.000 | 7.0386 | 2.0993 | 0.0216 | 0.9170 |

Table 7: Performance of FedBARRE on FMNIST under different privacy levels ($P$) with $M = 1$ classifier. Utility metric ($U$) is accuracy, and privacy metrics ($P$) include PSNR, MSE, and SSIM.

| $P$ | PSNR ($\downarrow$,$P$) | MSE ($\uparrow$,$P$) | SSIM ($\downarrow$,$P$) | Test-Acc ($\uparrow$,$U$) |
|---|---|---|---|---|
| 0.100 | 5.6992 | 2.1770 | 0.0233 | 0.7936 |
| 0.200 | 5.7425 | 2.1594 | 0.0222 | 0.7724 |
| 0.300 | 5.7995 | 2.1362 | 0.0188 | 0.7300 |
| 0.400 | 5.7889 | 2.1362 | 0.0200 | 0.7640 |
| 0.500 | 5.7780 | 2.1438 | 0.0208 | 0.7746 |
| 0.600 | 5.8490 | 2.1063 | 0.0201 | 0.7866 |
| 0.700 | 5.7774 | 2.1399 | 0.0204 | 0.7808 |
| 0.800 | 5.8873 | 2.0908 | 0.0210 | 0.7896 |
| 0.900 | 5.7902 | 2.1373 | 0.0226 | 0.7770 |
| 1.000 | 5.8768 | 2.0962 | 0.0233 | 0.7650 |

Table 8: Performance of FedBARRE on FMNIST under different privacy levels ($P$) with $M = 5$ classifiers. Utility metric ($U$) is Test-Acc, and privacy metrics ($P$) include PSNR, MSE, and SSIM.

| $P$ | PSNR ($\downarrow$,$P$) | MSE ($\uparrow$,$P$) | SSIM ($\downarrow$,$P$) | Test-Acc ($\uparrow$,$U$) |
|---|---|---|---|---|
| 0.100 | 5.7862 | 2.1369 | 0.0281 | 0.7272 |
| 0.200 | 5.8371 | 2.1102 | 0.0248 | 0.7266 |
| 0.300 | 5.7345 | 2.1578 | 0.0289 | 0.7514 |
| 0.400 | 5.8302 | 2.1101 | 0.0241 | 0.7190 |
| 0.500 | 5.8428 | 2.1078 | 0.0233 | 0.7422 |
| 0.600 | 5.8738 | 2.0941 | 0.0243 | 0.6872 |
| 0.700 | 5.8248 | 2.1141 | 0.0247 | 0.7510 |
| 0.800 | 5.9126 | 2.0781 | 0.0261 | 0.7432 |
| 0.900 | 6.0388 | 2.0195 | 0.0223 | 0.7240 |
| 1.000 | 6.0548 | 2.0119 | 0.0225 | 0.6960 |

Table 9: Performance of FedBARRE on FMNIST under different privacy levels ($P$) with $M = 9$ classifiers. Utility metric ($U$) is Test-Acc, and privacy metrics ($P$) include PSNR, MSE, and SSIM.

| $P$ | PSNR ($\downarrow$,$P$) | MSE ($\uparrow$,$P$) | SSIM ($\downarrow$,$P$) | Test-Acc ($\uparrow$,$U$) |
|---|---|---|---|---|
| 0.100 | 6.2521 | 1.9224 | 0.0236 | 0.7590 |
| 0.200 | 6.1289 | 1.9749 | 0.0253 | 0.7546 |
| 0.300 | 6.1411 | 1.9755 | 0.0240 | 0.7424 |
| 0.400 | 6.2428 | 1.9288 | 0.0241 | 0.7344 |
| 0.500 | 6.1301 | 1.9741 | 0.0231 | 0.7578 |
| 0.600 | 6.1436 | 1.9701 | 0.0265 | 0.7470 |
| 0.700 | 6.2486 | 1.9272 | 0.0279 | 0.7736 |
| 0.800 | 6.1916 | 1.9512 | 0.0259 | 0.7650 |
| 0.900 | 6.2785 | 1.9146 | 0.0270 | 0.7420 |
| 1.000 | 6.3353 | 1.8911 | 0.0227 | 0.7540 |

Table 10: Performance of FedBARRE on CIFAR-10 under different privacy levels ($P$) with $M = 1$ classifier. Utility metric ($U$) is Test-Acc, and privacy metrics ($P$) include PSNR, MSE, and SSIM.

| $P$ | PSNR ($\downarrow$,$P$) | MSE ($\uparrow$,$P$) | SSIM ($\downarrow$,$P$) | Test-Acc ($\uparrow$,$U$) |
|---|---|---|---|---|
| 0.100 | 6.2060 | 3.8926 | 0.0307 | 0.4212 |
| 0.200 | 6.4177 | 3.7046 | 0.0305 | 0.4058 |
| 0.300 | 6.3902 | 3.7336 | 0.0277 | 0.4944 |
| 0.400 | 6.4900 | 3.6410 | 0.0292 | 0.4894 |
| 0.500 | 6.6240 | 3.5432 | 0.0319 | 0.4044 |
| 0.600 | 6.6312 | 3.5197 | 0.0321 | 0.4690 |
| 0.700 | 6.7394 | 3.4345 | 0.0261 | 0.4134 |
| 0.800 | 6.7635 | 3.4142 | 0.0278 | 0.4910 |
| 0.900 | 6.8886 | 3.3178 | 0.0334 | 0.4974 |
| 1.000 | 6.9028 | 3.3220 | 0.0261 | 0.4000 |

Table 11: Performance of FedBARRE on CIFAR-10 under different privacy levels ($P$) with $M = 5$ classifiers. Utility metric ($U$) is Test-Acc, and privacy metrics ($P$) include PSNR, MSE, and SSIM.

| $P$ | PSNR ($\downarrow$,$P$) | MSE ($\uparrow$,$P$) | SSIM ($\downarrow$,$P$) | Test-Acc ($\uparrow$,$U$) |
|---|---|---|---|---|
| 0.100 | 6.3794 | 3.7342 | 0.0329 | 0.4018 |
| 0.200 | 6.3811 | 3.7092 | 0.0303 | 0.4596 |
| 0.300 | 6.5300 | 3.6106 | 0.0318 | 0.4592 |
| 0.400 | 6.4526 | 3.6515 | 0.0270 | 0.4364 |
| 0.500 | 6.4795 | 3.6352 | 0.0286 | 0.4330 |
| 0.600 | 6.7097 | 3.4444 | 0.0313 | 0.4746 |
| 0.700 | 6.5981 | 3.5309 | 0.0274 | 0.4996 |
| 0.800 | 6.7497 | 3.4141 | 0.0300 | 0.4676 |
| 0.900 | 6.5248 | 3.5772 | 0.0273 | 0.4024 |
| 1.000 | 6.9034 | 3.3012 | 0.0320 | 0.4228 |

Table 12: Performance of FedBARRE on CIFAR-10 under different privacy levels ($P$) with $M = 9$ classifiers. Utility metric ($U$) is Test-Acc, and privacy metrics ($P$) include PSNR, MSE, and SSIM.

| $P$ | PSNR ($\downarrow$,$P$) | MSE ($\uparrow$,$P$) | SSIM ($\downarrow$,$P$) | Test-Acc ($\uparrow$,$U$) |
|---|---|---|---|---|
| 0.100 | 6.2901 | 3.8298 | 0.0300 | 0.4822 |
| 0.200 | 6.3301 | 3.7835 | 0.0292 | 0.4872 |
| 0.300 | 6.3140 | 3.7918 | 0.0311 | 0.4566 |
| 0.400 | 6.3486 | 3.7462 | 0.0306 | 0.4964 |
| 0.500 | 6.5611 | 3.5791 | 0.0324 | 0.4510 |
| 0.600 | 6.5016 | 3.6189 | 0.0273 | 0.5216 |
| 0.700 | 6.5434 | 3.5916 | 0.0288 | 0.4452 |
| 0.800 | 6.6721 | 3.4667 | 0.0299 | 0.4784 |
| 0.900 | 6.7250 | 3.4211 | 0.0317 | 0.4902 |
| 1.000 | 6.7893 | 3.3745 | 0.0302 | 0.5098 |

| $\epsilon$ | PSNR ($\downarrow$, $P$) | MSE ($\uparrow$, $P$) | SSIM ($\downarrow$, $P$) | Test-Acc ($\uparrow$, $U$) |
|---|---|---|---|---|
| 0.050 | 9.0890 | 1.4209 | 0.0774 | 0.5190 |
| 0.100 | 9.0909 | 1.4172 | 0.0790 | 0.6180 |
| 0.150 | 9.0640 | 1.4333 | 0.0783 | 0.6816 |
| 0.200 | 9.0631 | 1.4301 | 0.0806 | 0.7222 |
| 0.250 | 9.2069 | 1.3880 | 0.0852 | 0.7536 |
| 0.300 | 9.2491 | 1.3859 | 0.0871 | 0.7740 |
| 0.350 | 9.1232 | 1.3952 | 0.0783 | 0.7928 |
| 0.400 | 9.1886 | 1.3938 | 0.0845 | 0.8082 |
| 0.450 | 9.1817 | 1.3881 | 0.0839 | 0.8224 |
| 0.500 | 9.1888 | 1.4061 | 0.0856 | 0.8338 |
| 0.550 | 9.0993 | 1.4057 | 0.0864 | 0.8420 |
| 0.600 | 8.9858 | 1.4204 | 0.0769 | 0.8516 |
| 0.650 | 9.2335 | 1.3710 | 0.0864 | 0.8568 |
| 0.700 | 9.2260 | 1.4007 | 0.0949 | 0.8614 |
| 0.750 | 9.0243 | 1.4054 | 0.0774 | 0.8646 |
| 0.800 | 9.1727 | 1.3781 | 0.0799 | 0.8666 |
| 0.850 | 9.1700 | 1.3833 | 0.0825 | 0.8700 |
| 0.900 | 9.1124 | 1.4100 | 0.0814 | 0.8734 |
| 0.950 | 9.2612 | 1.3772 | 0.0863 | 0.8774 |
| 1.000 | 9.1401 | 1.3840 | 0.0798 | 0.8804 |

Table 13: Performance of DP-Gas on MNIST under varying privacy budgets $\epsilon$. Privacy metrics (PSNR, MSE, SSIM) and utility metric (Test Accuracy) are reported. Arrows indicate whether a higher or lower value is desirable for privacy ($P$) or utility ($U$).

| $\epsilon$ | PSNR ($\downarrow$, $P$) | MSE ($\uparrow$, $P$) | SSIM ($\downarrow$, $P$) | Test-Acc ($\uparrow$, $U$) |
|---|---|---|---|---|
| 0.050 | 8.6251 | 1.2123 | 0.0977 | 0.3526 |
| 0.100 | 8.6420 | 1.2090 | 0.0957 | 0.4818 |
| 0.150 | 8.6350 | 1.2084 | 0.0971 | 0.5552 |
| 0.200 | 8.7630 | 1.1922 | 0.1061 | 0.5936 |
| 0.250 | 8.5778 | 1.2327 | 0.0983 | 0.6198 |
| 0.300 | 8.6198 | 1.2231 | 0.1000 | 0.6372 |
| 0.350 | 8.5708 | 1.2393 | 0.1004 | 0.6500 |
| 0.400 | 8.5446 | 1.2394 | 0.0962 | 0.6598 |
| 0.450 | 8.5802 | 1.2329 | 0.0987 | 0.6658 |
| 0.500 | 8.5509 | 1.2377 | 0.0965 | 0.6738 |
| 0.550 | 8.5690 | 1.2326 | 0.0981 | 0.6810 |
| 0.600 | 8.5448 | 1.2364 | 0.0964 | 0.6890 |
| 0.650 | 8.5107 | 1.2418 | 0.0936 | 0.6920 |
| 0.700 | 8.5777 | 1.2325 | 0.0971 | 0.6952 |
| 0.750 | 8.5172 | 1.2461 | 0.0958 | 0.7010 |
| 0.800 | 8.5379 | 1.2415 | 0.0979 | 0.7072 |
| 0.850 | 8.5409 | 1.2395 | 0.0970 | 0.7106 |
| 0.900 | 8.5452 | 1.2388 | 0.0962 | 0.7146 |
| 0.950 | 8.6016 | 1.2303 | 0.1005 | 0.7182 |
| 1.000 | 8.5630 | 1.2341 | 0.0954 | 0.7200 |

Table 14: Performance of DP-Gas on FMNIST under varying privacy budgets $\epsilon$. Privacy metrics (PSNR, MSE, SSIM) and utility metric (Test Accuracy) are reported. Arrows indicate whether a higher or lower value is desirable for privacy ($P$) or utility ($U$).

| $\epsilon$ | PSNR ($\downarrow$, $P$) | MSE ($\uparrow$, $P$) | SSIM ($\downarrow$, $P$) | Test-Acc ($\uparrow$, $U$) |
|---|---|---|---|---|
| 0.050 | 8.4177 | 1.5368 | 0.0375 | 0.1018 |
| 0.100 | 8.3542 | 1.5588 | 0.0346 | 0.0998 |
| 0.150 | 8.4087 | 1.5402 | 0.0342 | 0.0942 |
| 0.200 | 8.4083 | 1.5432 | 0.0343 | 0.0980 |
| 0.250 | 8.4384 | 1.5320 | 0.0330 | 0.1048 |
| 0.300 | 8.4445 | 1.5310 | 0.0333 | 0.0928 |
| 0.350 | 8.4867 | 1.5170 | 0.0381 | 0.0950 |
| 0.400 | 8.5047 | 1.5097 | 0.0386 | 0.1002 |
| 0.450 | 8.5626 | 1.4926 | 0.0404 | 0.1702 |
| 0.500 | 8.5703 | 1.4928 | 0.0398 | 0.1024 |
| 0.550 | 8.7314 | 1.4477 | 0.0408 | 0.2760 |
| 0.600 | 8.5948 | 1.4919 | 0.0423 | 0.4994 |
| 0.650 | 8.6170 | 1.4847 | 0.0446 | 0.5608 |
| 0.700 | 8.7251 | 1.4585 | 0.0454 | 0.5968 |
| 0.750 | 8.6231 | 1.4939 | 0.0484 | 0.7494 |
| 0.800 | 8.8274 | 1.4671 | 0.0583 | 0.8278 |
| 0.850 | 9.0016 | 1.4176 | 0.0753 | 0.8872 |
| 0.900 | 9.0128 | 1.4173 | 0.0699 | 0.9262 |
| 0.950 | 8.5827 | 1.4991 | 0.0573 | 0.9542 |

Table 15: Performance of PPFA on MNIST under varying privacy budgets $\epsilon$. Privacy metrics (PSNR, MSE, SSIM) and utility metric (Test Accuracy) are reported. Arrows indicate whether a higher or lower value is desirable for privacy ($P$) or utility ($U$).

| $\epsilon$ | PSNR ($\downarrow$, $P$) | MSE ($\uparrow$, $P$) | SSIM ($\downarrow$, $P$) | Test-Acc ($\uparrow$, $U$) |
|---|---|---|---|---|
| 0.050 | 7.2686 | 1.5346 | 0.0350 | 0.1062 |
| 0.100 | 7.4389 | 1.4706 | 0.0325 | 0.1254 |
| 0.150 | 7.4342 | 1.4734 | 0.0323 | 0.1078 |
| 0.200 | 7.3688 | 1.4969 | 0.0309 | 0.1162 |
| 0.250 | 7.3328 | 1.5094 | 0.0312 | 0.1046 |
| 0.300 | 7.3678 | 1.4970 | 0.0326 | 0.2064 |
| 0.350 | 7.4643 | 1.4642 | 0.0331 | 0.1906 |
| 0.400 | 7.5194 | 1.4509 | 0.0353 | 0.2180 |
| 0.450 | 7.5548 | 1.4345 | 0.0352 | 0.3632 |
| 0.500 | 7.5059 | 1.4501 | 0.0354 | 0.3834 |
| 0.550 | 7.6778 | 1.4007 | 0.0359 | 0.3964 |
| 0.600 | 7.5776 | 1.4295 | 0.0359 | 0.4804 |
| 0.650 | 7.6893 | 1.3969 | 0.0408 | 0.5282 |
| 0.700 | 7.9313 | 1.3309 | 0.0525 | 0.5822 |
| 0.750 | 8.0543 | 1.3043 | 0.0575 | 0.6362 |
| 0.800 | 8.2090 | 1.2810 | 0.0739 | 0.6814 |
| 0.850 | 8.3699 | 1.2773 | 0.0862 | 0.7274 |
| 0.900 | 8.2485 | 1.2968 | 0.0777 | 0.7938 |
| 0.950 | 8.3896 | 1.2896 | 0.1178 | 0.8236 |

Table 16: Performance of PPFA on FMNIST under varying privacy budgets $\epsilon$. Privacy metrics (PSNR, MSE, SSIM) and utility metric (Test Accuracy) are reported. Arrows indicate whether a higher or lower value is desirable for privacy ($P$) or utility ($U$).