# OpenReview forum: "FedBARRE: Privacy–Utility Optimized Perturbation Ensemble against Gradient Leakage Attacks in Federated Learning"
_ICLR.cc/2026/Conference — Submitted to ICLR 2026_

### Official Review · Reviewer_EY9r · 2025-10-27

**Soundness:** 2
**Presentation:** 3
**Contribution:** 2
**Rating:** 2
**Confidence:** 4

**Summary:**

The paper introduces FedBARRE, a novel FL framework to address the privacy-utility trade-off. The proposed method maps the privacy-utility trade-off to an adversarial training objective and leverages min-max optimization to optimize it. Extensive experimental results show that the proposed method effectively addresses the trade-off and defends against reconstruction attacks.

**Strengths:**

- The paper conducts extensive experiments to highlight the advantages of the proposed method.
- The proposed method is clear and straightforward for implementation.
- The paper is well-written

**Weaknesses:**

- Lack of theoretical analysis for the privacy guarantee.
- Some experiments have vague and suspicious results. For instance, $\epsilon$ is never defined and indicates how it's computed, but is still used to present the results.
- The privacy protection mechanism is not sufficient. Specifically, the data is used throughout every step of the optimization process without any protection. Even the noise is optimized with the data and becomes deterministic.

**Questions:**

- For the definition of $\rho(f)$, why the loss consider argument minimizaition by the noise $\delta$? Doesn't the noise $\delta = 0$ minimize this loss? If not, then $\delta$ is mapping $x$ to some other features?
- For $G(\alpha, \sigma)$, why is it convex w.r.t $\sigma$? How is this $G$ function elaborated throughout the paper? If $f$ is not convex, how do we ensure the structural property?
- How did you compute $\epsilon$ for your mechanism?
- Can the proposed method defend against Membership inference attacks?

---

### Official Review · Reviewer_HxEv · 2025-10-31

**Soundness:** 2
**Presentation:** 1
**Contribution:** 2
**Rating:** 2
**Confidence:** 3

**Summary:**

The submission tackles the issue of preserving privacy in federated learning, where the private data of the clients can be obtained by an adversary via gradient inversion attacks. The submission introduces a FedBARRE, a training algorithm in which clients learn optimized perturbations to their data (bounded within some min/max L2 norm) that minimizes the performance loss due to the perturbation. At each step, each client in the selected batch optimizes their perturbation to minimize their loss, followed by a model update to their local model. After each model update, the client tests the model on their local validation set and eventually sends the gradient of the model (among those encountered throughout this training) that minimizes this validation loss to the central server for model update aggregation. The authors test their method against other privacy-preserving methods from literature to demonstrate that FedBARRE achieves a better utility-privacy tradeoff.

**Strengths:**

1) The problem of achieving better privacy–utility tradeoff in FL is well-motivated and an important one.

2) The experiment covers different datasets and also compares their method to various existing methods.

3) The algorithms are easy to understand and intuitive.

**Weaknesses:**

1) There appears to be a fundamental disconnect between what the submission is claiming to do and what the FedBARRE algorithm actually does. First and foremost, the introduction presentes FedBARRER as a method using Randomized Ensemble
Classifiers and section 4.2 clearly labels the the objective ("Our final optimization goal...") as learning M model parameters (for each model in the ensemble) and the ensemble weights $\alpha$, but this is not what FedBARRE does: there is no computation of any ensemble weights, only a single final global model that results from a standard aggregation of the gradients sent by the calendars. In fact the optimization objective $\rho(\alpha)$ never brought up again after being briefly mentioned in the start of Section 5, where it is vaguely stated that the algorithm is "grounded in the optimization objective introduced in Section 4" with no justification (in particular it never comes up, implicitly or explicitly, in the actual algorithms). In fact, calling the models {$\theta_1,...,\theta_M$} in Algorithm 2 an "ensemble" is a stretch by itself, since each model is simply a clone of the previous model that continues with the training, so the "ensemble" component of the algorithm simply boils down to the client training a single model and checking its validation loss (using a local validation data that is never formally defined) at M points during the training, and then using epoch among these M checkpoints that achieves minimum error.

Further, the first key contribution listed in Section  is "We introduce and analyze three risk measures—standard risk, adversarial risk, and REC adversarial risk—and prove the convexity of the REC adversarial risk, establishing a solid theoretical foundation for ensemble optimization in federated learning." However, the terms "adversarial risk" or "REC adversarial risk" is never brought up again (Definitions 2 and 3 which introduce Privacy–Utility Risk and REC Privacy–Utility Risk do not have an adversarial component) until the conclusion, where it is again claimed that "We began by proving the convexity of the adversarial risk for the Randomized Ensemble Classifierand deriving a rigorous privacy–utility frontier, thereby providing provable privacy guarantees." There also seems to be no such theoretical analysis/proof; the claims made in Section 3.4 (and Appendix B) are relatively straightforward and does not result in a "rigorous privacy–utility frontier" as claimed.

2) In addition to the more serious concerns listed above, there are other presentation issues that interrupt the flow of the paper and decrease readability. Some examples include:
- The datasets for each client $|\mathcal{D}^{(k)}|$ or their "local validation sets" used during training are never properly defined.
- $\rho(\alpha)$ is repetitively defined __three__ times using the same full-line equation (Section 3.3, Section 4.2, Section 5) and is never brought up again. In fact, the only other time $\rho$ appears is when is it overloaded to denote the learning rate in Algorithm 2.
- The "privacy budget" $\epsilon$ used in almost all experiments comes out of the blue and is never defined. Only in section 6.4 do we -learn that smaller values of $\epsilon$ corresponds to larger perturbation radius, which is unclear why this is the cae.
- Some tables and figures, such as Figure 4 are hard to read / interpret. While the text claims that " a consistent trend emerges across all datasets" in Figure 4, the subsequently described trend is not at all easy to see from the figure, since the lines clearly demonstrate non-monotonicities and the scale is hard to read.

3) While the experiments compare FedBARRE to other methods using different "ensemble sizes" and "privacy budgets", they still use a single attack and do not test for different parameters of the FL settings (number of clients etc) or the parameters of the other methods, which brings into question the robustness of the utility/privacy gain achieved by their method in different settings.

**Questions:**

Can you address the inconsistencies listed in item (1) in the weaknesses above?

---

### Official Review · Reviewer_FhJw · 2025-11-01

**Soundness:** 2
**Presentation:** 2
**Contribution:** 2
**Rating:** 2
**Confidence:** 4

**Summary:**

The paper proposes FedBARRE to address gradient inversion attacks in Federated Learning (FL).
The core of FedBARRE is an ensemble of classification heads where input data is perturbed with randomized noise.
Experiments demonstrate that FedBARRE achieves better performance.

**Strengths:**

* The paper is generally well-structured and easy to follow.

* The experimental results demonstrate that the proposed method is effective in achieving a better privacy-utility balance against DLG.

**Weaknesses:**

* The core privacy contribution of Randomized Ensemble Classifier seems to stem from introducing randomness rather than a robust privacy mechanism. This type of random defense has been previously utilized in the literature to obfuscate gradients and deter deterministic reconstruction attacks [1]. The paper lacks a convincing analysis of its resilience against adaptive attacks [2].

* The objective function, specifically the inner minimization, is questionable. $\delta$ is optimized to minimize the loss, which means finding a perturbation that makes the model perform better on the perturbed data $(x+\delta)$. This formulation suggests that the model parameters are optimal to $(x+\delta)$.

* The paper lacks formal privacy or utility proofs typically required for defensive techniques.

* The experimental evaluation is not sufficiently comprehensive. It lacks comparison against several state-of-the-art methods [3,4]. Moreover, the paper omits any analysis of the computational cost and communication overhead.

[1] Precode - a generic model extension to preventdeep gradient leakage

[2] Bayesian Framework for Gradient Leakage

[3] See through gradients: Image batch recovery via gradinversion

[4] Refiner: Data refining against gradient leakage attacks in federated learning

**Questions:**

See Weaknesses.

---

### Official Review · Reviewer_LwnY · 2025-11-01

**Soundness:** 2
**Presentation:** 3
**Contribution:** 2
**Rating:** 2
**Confidence:** 4

**Summary:**

The paper addresses gradient leakage in horizontal FL with a semi‑honest server observing per‑client gradients. FedBARRE trains, per mini‑batch, an ensemble of M local classifiers under PGD‑bounded benign perturbations; each client mixes clean/perturbed losses, validates to pick the best submodel, and uploads only that submodel’s gradient, while the server aggregates per mini‑batch (FedSGD‑like). Experiments on MNIST, FMNIST, and CIFAR‑10/100 use 4 IID clients; attacks are simulated using Inverting Gradients (IG), and DLG is executed on the first client’s gradients during rounds 9–11. Utility is test accuracy; privacy uses MSE/PSNR/SSIM; baselines include standard local DP mechanisms (e.g., Wei et al., 2021), PPFA, and Noise‑Add; ablations vary ensemble size M and a privacy budget with privacy–utility curves.

**Strengths:**

1. Implementable pipeline: Clear workflow and pseudocode (Figure 1; Algorithms 1–2) make adoption in FedSGD‑style systems straightforward.
2. Practical measurement: Uses interpretable leakage metrics (MSE/PSNR/SSIM) and presents privacy–utility fronts (e.g., Figure 3).
3. Tunable knobs with ablations: Documents effects of ensemble size M and privacy budget on privacy/utility (Section 6.3 Table 3; Section 6.4 Figure 4).

**Weaknesses:**

1. Theory–implementation gap: Section 4.2 optimizes ensemble weights (α) but Algorithm 2 implements deterministic best‑of‑M selection without learning/sampling α.
2. Guarantee wording inconsistency: The paper says “not a formal privacy guarantee” (Sec 3.4) while the Conclusion claims “provable privacy guarantees.”
3. Narrow evaluation and no systems costs: Small IID setup (4 clients, 30 rounds), IG (with DLG executed in rounds 9–11) on a single client, no secure‑aggregation/aggregated‑only setting, and no compute/comm/memory/energy reporting.

**Questions:**

1. How will α be aligned with the implementation? Will you implement learning/sampling of α (e.g., with entropy/temperature) or formally justify best‑of‑M as the intended one‑hot solution and revise Section 4.2 accordingly?
2. Can you broaden and quantify the evaluation? Add non‑IID and larger‑K runs, an aggregated‑only (secure aggregation) scenario, and report cost curves versus M/PGD steps/privacy budget.

---

### Meta-Review · Area_Chair_Q2fp · 2026-01-02

**Summary:**

This paper proposes a federated learning framework that aims to mitigate gradient-based data reconstruction attacks. While reviewers found the problem setting interesting, they raised concerns about the clarity and alignment of the optimization objective with the implemented algorithm, the absence of formal privacy guarantees, and the limited scope of the empirical evaluation. In the absence of an author response, these issues remain unresolved, and hence the AC, sadly, recommends ``rejection``.

**Reviewer Concerns:**

Reviewer **HxEv** raised concerns about a mismatch between the stated optimization objective and the implemented algorithm. Reviewer **LwnY** similarly noted a theory–implementation gap and the narrow experimental setup. Reviewers **FhJw** and **EY9r** questioned the lack of formal privacy guarantees, and the limited evaluation against more sophisticated attacks. As no author response was provided, these concerns remain unaddressed.

**Reviewer Scores:**

In the absence of any author response, the AC does not expect any reviewer to revise their assessment.

---

### Decision · Program_Chairs · 2026-01-26

Reject